# Prevalence of the *Puumala orthohantavirus* Strains in the Pre-Kama Area of the Republic of Tatarstan, Russia

**DOI:** 10.3390/pathogens9070540

**Published:** 2020-07-06

**Authors:** Yuriy Davidyuk, Anton Shamsutdinov, Emmanuel Kabwe, Ruzilya Ismagilova, Ekaterina Martynova, Alexander Belyaev, Eduard Shuralev, Vladimir Trifonov, Tatiana Savitskaya, Guzel Isaeva, Svetlana Khaiboullina, Albert Rizvanov, Sergey Morzunov

**Affiliations:** 1OpenLab “Gene and Cell Technologies”, Kazan Federal University, Institute of Fundamental Medicine and Biology, Kazan 420008, Russia; davi.djuk@mail.ru (Y.D.); shamsutdinov2006@yandex.com (A.S.); emmanuelkabwe@ymail.com (E.K.); ignietferro.venivedivici@gmail.com (E.M.); rizvanov@gmail.com (A.R.); 2Kazan Research Institute of Epidemiology and Microbiology, Kazan 420012, Russia; vatrifonov@mail.ru (V.T.); tatasav777@mail.ru (T.S.); guisaeva@rambler.ru (G.I.); 3OpenLab “Omics Technology”, Institute of Fundamental Medicine and Biology, Kazan Federal University, Kazan 420008, Russia; ruz-ismagilova@yandex.ru; 4Department of Zoology and General Biology, Zoological Museum, Institute of Fundamental Medicine and Biology, Kazan Federal University, Kazan 420008, Russia; crocidura@mail.ru; 5Department of Applied Ecology, Institute of Environmental Sciences, Kazan Federal University, Kazan 420097, Russia; eduard.shuralev@mail.ru; 6Medical Academy of the Ministry of Health of the Russian Federation, Kazan 420012, Russia; 7Department of Microbiology and Immunology, University of Nevada, Reno, NV 89557, USA; 8Department of Pathology, University of Nevada, Reno, NV 89557, USA

**Keywords:** *Puumala orthohantavirus*, genetic diversity, phylogenetic analysis, *Myodes glareolus*

## Abstract

*Puumala orthohantavirus* (PUUV) causes nephropathia epidemica (NE), a mild form of hemorrhagic fever with renal syndrome (HFRS) commonly diagnosed in Europe. The majority of HFRS cases in the European part of Russia are diagnosed in the Volga Federal District, which includes the Republic of Tatarstan (RT). The current study aims to analyze the genetic variability of PUUV in Pre-Kama region of the RT bounded by the Volga, Kama, and Vyatka rivers. In 2017, bank voles were caught in seven isolated forest traps in the Pre-Kama region and for the 26 PUUV-positive samples, the partial small (S), medium (M), and large (L) genome segment sequences were obtained and analyzed. It was determined that all identified PUUV strains belong to the Russian (RUS) genetic lineage; however, the genetic distance between strains is not directly correlated with the geographical distance between bank vole populations. One of the identified strains has S and L segments produced from one parental strain, while the M segment was supplied by another, suggesting that this strain could be the reassortant. We suggest that the revealed pattern of the PUUV strains distribution could be the result of a series of successive multidirectional migratory flows of the bank voles to the Pre-Kama region in the postglacial period.

## 1. Introduction

Orthohantaviruses (genus *Orthohantavirus*, family Hantaviridae, order Bunyavirales) are zoonotic pathogens circulating in their respective natural reservoirs in the Old and New Worlds [1]. There are two significant groups of orthohantaviruses associated with the unique clinical presentation of infection. In South and North America, orthohantavirus infection is diagnosed as a cardiopulmonary syndrome (HCPS), where cardiovascular and pulmonary dysfunction are commonly identified [1]. In contrast, in Eurasia, disturbed blood coagulation and kidney insufficiency are noted in patients, a condition referred to as hemorrhagic fever with renal syndrome (HFRS) [2]. Annually, thousands of orthohantavirus infection cases are recorded in the world [3], making it a serious health threat.

Orthohantavirus virions are spherically shaped containing a segmented RNA genome of the negative polarity. The small (S), medium (M), and large (L) segments encode nucleocapsid (N) protein, envelope glycoproteins Gn and Gc, and RNA-dependent RNA polymerase (RdRp), respectively [4]. In some species, the S segment also encodes a small non-structural (NSs) protein [5]. It is believed that each hantavirus is associated with one or more species of mammals serving as its natural hosts [6]. Therefore, the epidemiology of orthohantavirus infections significantly depends on factors affecting the prevalence of their hosts.

*Puumala orthohantavirus* (PUUV) causes nephropathia epidemica (NE), a mild form of HFRS commonly diagnosed in Europe [7]. The natural carriers of PUUV are bank voles (*Myodes glareolus*), whose range includes vast territories from Europe to Western Siberia. Large bank vole populations carrying multiple hantavirus strains could facilitate the emergence of novel genetically different virus strains. Currently, based on the data on the genetic variations in the S segment sequences, eight PUUV genetic lineages are identified in Eurasia: the Danish (DAN), north-Scandinavian (N-SCA), the south-Scandinavian (S-SCA), Central European (CE), Alpe-Adrian (ALAD), Latvian (LAT), Finnish (FIN) and the Russian (RUS) lineages [8]. The last two families were identified in Russia, with RUS lineage being found in the Volga region (Samara region, Republic of Udmurtia, Republic of Tatarstan (RT), Republic of Bashkortostan, Republic of Mordovia) [9], and FIN lineage being found in Karelia and Western Siberia [10].

In the European part of Russia, PUUV is identified as the leading cause of HFRS. The majority of HFRS cases are diagnosed in the Volga Federal District, which includes the Republic of Tatarstan (RT) [11]. Seven hundred and seventy-eight cases were reported in the RT in the first 10 months of 2019 [12]. A significant proportion of these cases was diagnosed in the Pre-Kama region of RT, which is located between the Volga and Kama rivers [13].

This region’s landscape includes forest-steppe with the presence of a significant number of isolated broad-leaved forests, serving as the habitat for the bank voles. Due to the long-term field studies in the Pre-Kama region, bank voles were found to be PUUV positive at the rate of 2.2–3.6% annually [14]. Isolated forest habitats could maintain isolated bank vole populations infected with genetically different PUUV strains.

In the previous investigations, we have performed a comparative analysis of partial S and M segment sequences of the PUUV strains circulating in the Pre-Kama and Trans-Kama area and have demonstrated a significant variation of PUUV strains’ genome in the RT [15,16]. It has been shown that the PUUV strains in the bank voles from Zelenodolsky and Vysokogorsky districts diverged significantly from the virus strains found in Laishevsky and Pestrechinsky districts [17]. The current study aims to analyze the genetic variability of PUUV in the Pre-Kama region. Also, we sought to identify PUUV strains, which could be the product of recombination and reassortment. These results could be used to map PUUV strains circulating in RT and investigate the mechanisms leading to the emergence of the new PUUV strains in RT.

## 2. Results and Discussion

In 2017, 119 bank voles were caught in seven isolated forest traps in the Pre-Kama region bounded by the Volga, Kama, and Vyatka rivers (Figure 1). PUUV RNA was detected in lung tissue samples of 29 (24.4%) animals using reverse transcription-polymerase chain reaction (RT-PCR). For the majority of PUUV-positive samples, the S, M, and L genome segments were amplified sizes 1057 bp, 1014 bp, and 665 bp, respectively (Table 1). Further, in text, the PUUV strains investigated in this study will be denoted as “RT-2017”.

Analysis of PUUV sequences obtained from animals captured in each trapping site revealed no significant differences between the nucleotide sequences of the S segments, less than 1% (Table 2). All obtained “RT-2017” sequences could be grouped into three clusters: cluster A, strains from S1–S5 sites with divergence ranging from 0.1 to 2.4%, cluster B, strains from site S6; and cluster C, strains from sites S7 and S8, where the variation was 2.2–2.3%. The “RT-2017” sequences’ divergence between different clusters was higher, reaching 5.2–6.1% (Table 2). Analysis of these and previously identified RUS lineage PUUV sequences from some regions of Russia (Samara_49/CG/2005, Puu/Kazan, CG1820, DTK/Ufa-97 strains) showed 4.4–6.7% divergence. At the same time, it was higher, exceeding 15%, when compared to sequences of FIN, CE, and N-SCA lineages (Table 3). We concluded that all “RT-2017” S segment belongs to the RUS genetic lineage.

Analysis of the amino acid (aa) sequences of amplified S segment fragments revealed that they were 100% identical within strains from each trapping site. When comparing “RT-2017” aa sequences and previous PUUV isolates, three aa substitutions were identified: Ile168Val, Arg242Lys, and Ile260Val (Figure 2). Ile168 was detected only in strains of the S1 site; Arg242 was found in isolates from S1, S6 sites and in Samara_49/CG/2005 strain, while other RUS lineage strains have Lys242; Val260 was found only in strains from S6 site and in Samara_49/CG/2005, CG1820 and DTK/Ufa-97 strains (Figure 2).

Analysis of the M segment sequences identified the difference between zero and 0.5% within “RT-2017” from each site (Table 4). Also, similar to that in S segment, three clusters were identified in M segment sequences: cluster A strains from sites S1–S5 (0.2–2.7% divergence); cluster B, strain only from site S6; cluster C, strains from sites S7 and S8 (2.1–2.3% divergence). The variation between sequences from different clusters was 5.8–8.5%, which was slightly higher than that in the S segment. When “RT-2017” strains were compared to strains from different genetic lineages, divergence was ranging from 6.3 to 8.7% from Samara_49/CG/2005 and Puu/Kazan, which belong to the RUS lineage. Interestingly, the difference between “RT-2017” and Baskiria strains (CG1820 and DTK/Ufa-97) sequences was higher (13.9–17.3%), although they also belong to the same RUS lineage. Variation in nucleotide sequences between “RT-2017” and FIN, CE, and N-SCA families was even higher (17.7–25.5%) (Table 3).

Differences in the aa sequences of the M segment within the clusters were 0.0–0.3% while it was higher, 0.3–1.2%, between sequences from different clusters (Table 4). When “RT-2017” aa sequences were compared to Samara_49/CG/2005 and Puu/Kazan strains, divergence was 0.3–1.2%; however, differences were more pronounced when compared to CG1820 and DTK/Ufa-97 strains, 3.0–3.6%, and 1.5–2.1%, respectively (Table 5). The number of aa substitutions specific for “RT-2017” obtained from the individual sites was identified. For example, the Ala521Val mutation was found in strains from the S8 site, while Ile726Val substitution was characteristic from the S6 site. Also, the Thr660Met mutation was detected in the strains from the S1 site. It should be noted that Ile577Val mutation was found in all strains from cluster A; however, all “RT-2017” contained Lys at aa 544 positions, similar to that in Puu/Kazan strain, not Arg, which is present in Samara_49/CG/2005 strain (Figure 3).

Similar to that in S and M segments, analysis of the L segment sequences revealed limited differences (0.0–0.8%) between “RT-2017” strains within each site. Also, sequences could be separated into the same three clusters: cluster A, S1–S5 sites (divergence 0.3–2.9%); cluster B, site S6 (two sequences are identical); cluster C, sites S7, and S8 (the difference between sites is 3.1%). Gaps between clusters A and C ranged from 4.0 to 7.0%. A similar degree of divergences was found between groups B and C (6.7%); however, more variations were observed between clusters A and B (6.5–8.1%) (Table 6). Analysis of sequences from different genetic lineages revealed that the differences between “RT-2017” and Samara_49/CG/2005 and Puu/Kazan were 6.7–8.6% and 6.7–8.9%, respectively. Much higher variations (14.3–15.9%) were found between “RT-2017” and strains from Bashkortostan (CG1820 and DTK/Ufa-97), which belong to the RUS lineage. In contrast, more differences were found between “RT-2017” and FIN, CE, and N-SCA lineages (17.6 to 24.7%) (Table 3). Additionally, specific aa substitutions were identified in “RT-2017” from S5 and S6 sites, Gln413His and Ser393Thr, respectively.

L segment aa sequences had limited differences within “RT-2017” (0.0–0.5%) (Table 6). Also, the divergences between “RT-2017” and previously identified RUS lineage PUUV Samara_49/CG/2005, Puu/Kazan and CG1820 + DTK/Ufa-97 were 0.9–1.4%, 0.5–0.9% and 1.4–1.8%, respectively (Table 5).

S and M segment phylogenetic trees have similar topology. “RT-2017” sequences on both trees were grouped into three subclades within the RUS clade: (A) sites S1–S5, (B) site S6, and (C) sites S7 and S8 (Figure 4, Figure 5, Appendix A). The most interesting is the location of branches corresponding to strains from sites S2 and S3. On the trees for partial S and L segments, the branch of the strain PUUV/Pestretsy/MG_1131/2017 (from site S2) is located separately from the sites S1, S3, S4, and S5 strains (Figure 4, Figure 6, Appendix A). In contrast, PUUV/Pestretsy/MG_1131/2017 (site S2) and PUUV/Lenino-Kokushkino/MG_1140/2017 (site S3) M segments are together with the strains from the S1 site (Figure 5 and Appendix A). Similar was the localization of the L segments on the phylogenetic tree with one exception, where the branch corresponding to the “RT-2017” from the S7 and S8 site was located closer to the subclade S1-S5, suggesting a closer relationship between these sequences (Figure 6 and Appendix A). It should be noted that on all three trees, the branch corresponding to strains CG1820 and DTK/Ufa-97 from Bashkortostan was located separately from the other strains of the RUS lineage, which could indicate the existence of a different PUUV sub-lineages in the Volga region.

The differences in the topology of phylogenetic trees for S, M, and L segments could be the result of the reassortment [18]. Based on the analysis of the nucleotide sequences, we have identified several genome variants in individual clusters (Table 7).

We have found variants of segments SB, MB, and LB present in cluster B, while variants SC, MC, and LC were identified in cluster C (subvariants C1 and C2 were found for strains from sites S7 and S8, respectively). Also, for five sites within the cluster A, there were three variants of the S segment, two variants of M segment and three variants of L segment sequences. In the sites, S4 and S5, identical options of segments SA3, MA3, and LA3 were identified, and in the S1 and S3, variants SA1, MA1, and LA1 segment were found. Additionally, a combination of genome segments SA2-MA1-LA2 could be identified in the PUUV/Pestretsy/MG_1131/2017 strain, suggesting that this strain could be a reassortant. It appears that this strain has S and L segments produced from one parental strain, while the M segment was supplied by another. This is similar to what has been previously shown for some naturally and in vitro-generated reassortants. For instance, in the genome of an interspecific reassortant obtained in vitro from the orthohantaviruses Andes (ANDV) and Sin Nombre (SNV), the S and L segments were inherited from SNV and the M segment from ANDV [19]. It should be noted, however, that a significant number of PUUV reassortants containing a combination of M and L segments coming from the same ancestral strain and the S segment from another were found in bank vole populations in Central Finland [20]. Also, possible natural-occurring PUUV reassortants containing S and M segments from different ancestor strains were found in Slovakia [21].

Our analysis confirms the high degree of homogeneity of PUUV genomes in a specific site and a significant genetic diversity between strains circulating in different locations. Also, the genetic distance between the “RT-2017” appears to be independent of the geographical distance between the bank vole habitats. On the one hand, the difference between the nucleotide sequences of the “RT-2017” from S4 and S5 sites, and from three other sites of cluster A located at a distance of 35–45 km is lower than with the strains from the S6 site, located 17–25 km apart (Figure 1, Table 2, Table 4, and Table 6). On the other hand, “RT-2017” from clusters B and C, separated by 140 km, have closer related M segment sequence as compared to strains from cluster A, geographically located between them (Figure 1, Table 4). Additionally, M segment from these clusters are located in the same subclade, labeled as ’SOUTH’, with strains circulating in Nizhnekamsky and Tukaevsky districts in the Trans-Kama area, suggesting a close relationship of these strains (Figure 5 and Appendix A).

This study included the part of the RT located between the Volga River on the West, Kama River on the South, and Vyatka River on the east, making some the “peninsula,” surrounded by three rivers. Currently, Volga and Kama rivers are an insurmountable obstacle for the bank voles. However, until the 1950s, these rivers had a lesser width, making colonization in these directions feasible. According to Deconenko et al. [10], the post-glacial recolonization of the Middle Volga by the bank vole occurred from south to north along the banks of the Volga. Based on this hypothesis, it could be suggested that PUUV strains were introduced into the Trans-Kama and Pre-Kama areas from the south, which explains the relatively small genetic distance between strains from the Trans-Kama area and clusters B and C from the Pre-Kama area. On the other hand, after recolonization, changes in climate, environmental, and anthropogenic factors could cause a secondary migration of the bank voles into the same area from north and/or northeast. In another study, [8] it was suggested that the movement of the bank voles to the Volga region occurred from the west, most likely from Latvia. Therefore, identified in this study distribution of PUUV genome variants in cluster A could be the result of one or more of these secondary migrations. In this case, modifications of the PUUV genomes in clusters B and C, commonly found in isolated forests, are believed to be preserved forms of the PUUV genome from the first wave of vole migrations.

The fact that PUUV strains in the RT can have different origins and can be the result of several waves of bank vole migrations could be confirmed by the results of phylogenetic analysis of the partial S segment sequences (Figure 7 and Appendix A).

PUUV strains from cluster A and the subclades’ Northwest of RT’, including strains circulating in the bank vole populations west and north of Kazan [17], appear genetically distant, although the geographical distance between these areas is in the range of 20–70 km. This suggests that a group of strains in the North-West of RT was also formed as a result of a separate migration wave of bank voles.

We believe that understanding the PUUV strain distribution requires a comprehensive study of the virus genome variants and the factors affecting the migration of bank voles. Particular attention should be given to the natural and physical boundaries, which could form the point of contact between different bank vole populations. This site of communication could be the location where the new PUUV genomes could be created, including potentially those more dangerous for humans.

## 3. Materials and Methods 

### 3.1. Bank Vole Trapping and Abundance

Trapping of bank vole was conducted in spring (April–May), summer (July) and fall (September–October) 2017. Information about the geographic locations of the trapping sites is shown in Figure 1. Standard mouse-type snap traps were set in lines of 50 and spaced 5 m apart. Traps were baited and left for one night. Further, small animals were identified by morphological characteristics following Pavlinov et al. [22] method and immediately frozen. Lung tissues were collected and used tor RNA extracting.

### 3.2. RNA extraction, cDNA synthesis and Polymerase Chain Reaction (PCR)

Total RNA was extracted from the lung tissues of bank voles with TRIzol Reagent (Invitrogen Life Technologies^TM^, Waltham, MA, USA), following the manufacturer’s recommendations. cDNA was synthesized using Thermo Scientific RevertAid Reverse Transcriptase (“Thermo Fisher Scientific”, Waltham, MA, USA). RT-PCR was undertaken using TaqPol polymerase kit (“Sileks”, Badenweiler, Germany). Primers used for RT-PCRs and sequencing analysis are summarized in Table 8.

PCR products were purified using Isolate II PCR and Gel Kit (“Bioline”, London, UK) and sequenced using ABI PRISM 310 big Dye Terminator 3.1 sequencing kit (ABI, Waltham, MA, USA). Sequences were deposited in the GenBank database under accession no. MT495363-MT495388 for partial S segment, MT495323-MT495344 and MT495352-MT495355 for partial M segment and MT502385-MT502410 for partial L segment.

### 3.3. Phylogenetic Analysis

For phylogenetic analysis, nucleotide sequences of PUUV strains obtained in this work and from GenBank were used. These included for segment S: Samara_49/CG/2005, AB433843; Puu/Kazan, Z84204; CG1820, M32750; DTK/Ufa-97, AB297665; Sotkamo 2009, HE801633; PUUV/Pieksamaki/human_lung/2008, JN831947; Mu/07/1219, KJ994776; PUUV/Ardennes/Mg156/2011, KT247592; PUUV/Orleans/Mg29/2010, KT247595; Umea/hu, AY526219, as well as the sequences obtained by us earlier in the same regions of the RT MG573266-MG573274, MG573276-MG573296, MG573299-MG573302 and sequences of 8 strains from Trans-Kama area MT472648 and MT495356-MT495362 [16]. Sequences for segment M analysis included: Samara_49/CG/2005, AB433850; Puu/Kazan, Z84205; CG1820, M29979; DTK/Ufa-97, AB297666; Sotkamo 2009, HE801634; PUUV/Pieksamaki/human_lung/2008, JN831948; Mu/07/1219, KJ994777; PUUV/Ardennes/Mg156/2011, KT247603; PUUV/Orleans/Mg29/2010, KT247601; Umea/hu, AY526218 and sequences of 7 strains from Trans-Kama area MT495345-MT495351 [16]. For segment L: Samara_49/CG/2005, AB574183; Puu/Kazan, EF405801; CG1820, KT885050; DTK/Ufa-97, AB297667; Sotkamo 2009, HE801635; PUUV/Pieksamaki/human_lung/2008, JN831949; Mu/07/1219, KJ994778; PUUV/Ardennes/Mg156/2011, KT247609; PUUV/Orleans/Mg29/2010, KT247605; Umea/hu, AY526217. As outgroup, sequences of Tula orthohantavirus AF164093, NC_005228 and NC_005226 for segments S, M and L, respectively, were used.

The nucleotide alignments and phylogenetic analysis of the PUUV strains were done using MegAlign program (Clustal W algorithm) located in the DNASTAR software package Lasergene (DNASTAR, Madison, WI, USA; https://www.dnastar.com/) and MEGA v6.0 [23]. Phylogenetic trees were constructed using maximum parsimony (MP) and maximum likelihood (ML) methods included in Mega v6.0. [23] The Tamura 3-Parameter model for all the three segments was used as an optimal substitution model in ML. The bootstrap values calculated for 1000 replicates are given in percentage and the values less than 70% are not shown. The tree is drawn to scale, with branch lengths calculated in the number of substitutions per site. For comparison reasons the MP-trees for all the segments are available in the Appendix A.

## 4. Conclusions

We analyzed partial S, M, and L segment sequences of PUUV circulating in the bank vole populations in Pre-Kama area of the RT. It was determined that all identified PUUV strains belong to the RUS genetic lineage; however, the genetic distance between strains is not directly correlated with the geographical distance between bank vole populations. We believe that the revealed pattern of the PUUV strains’ distribution could be the result of a series of successive multidirectional migratory flows of the bank voles to the Pre-Kama region in the postglacial period.

## Figures and Tables

**Figure 1 pathogens-09-00540-f001:**
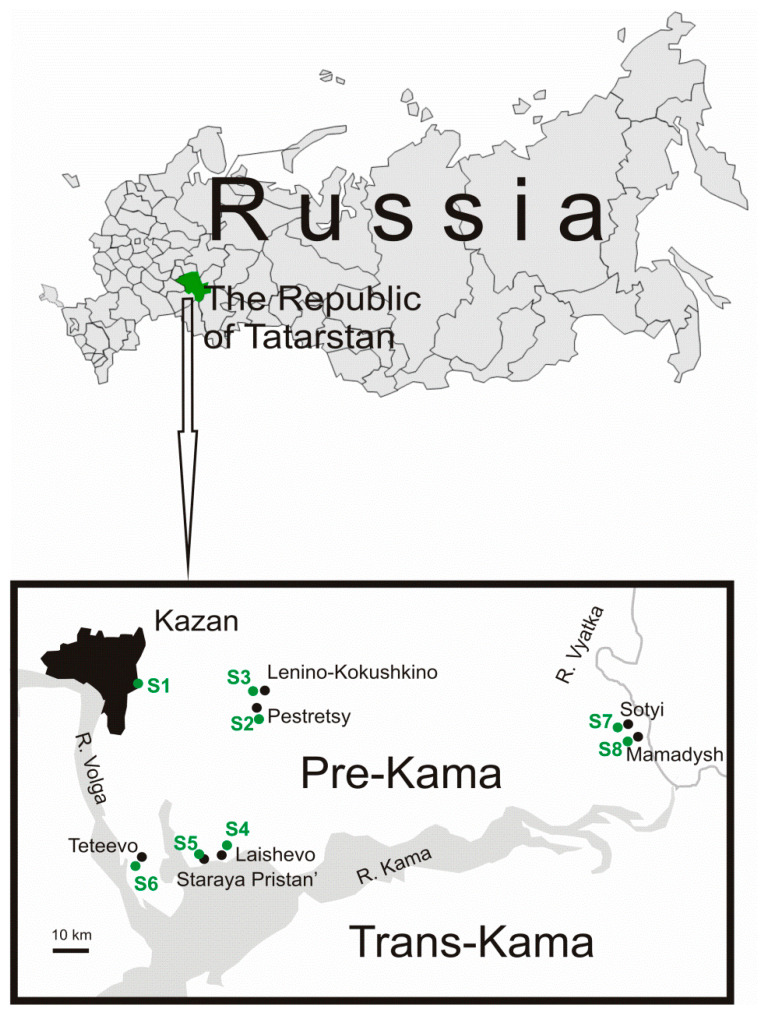
Trapping sites localization.

**Figure 2 pathogens-09-00540-f002:**
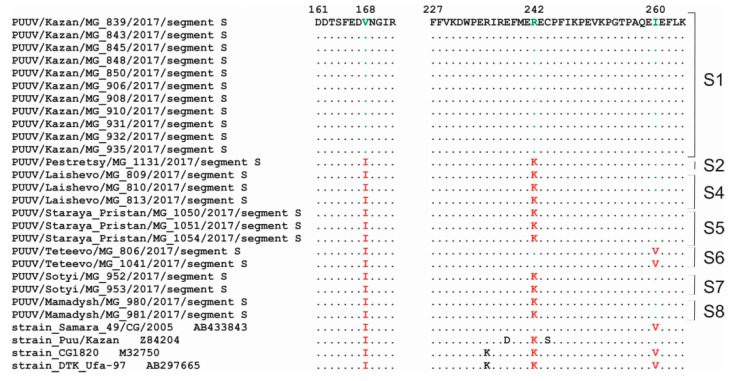
Mutations in the aa sequences of N protein.

**Figure 3 pathogens-09-00540-f003:**
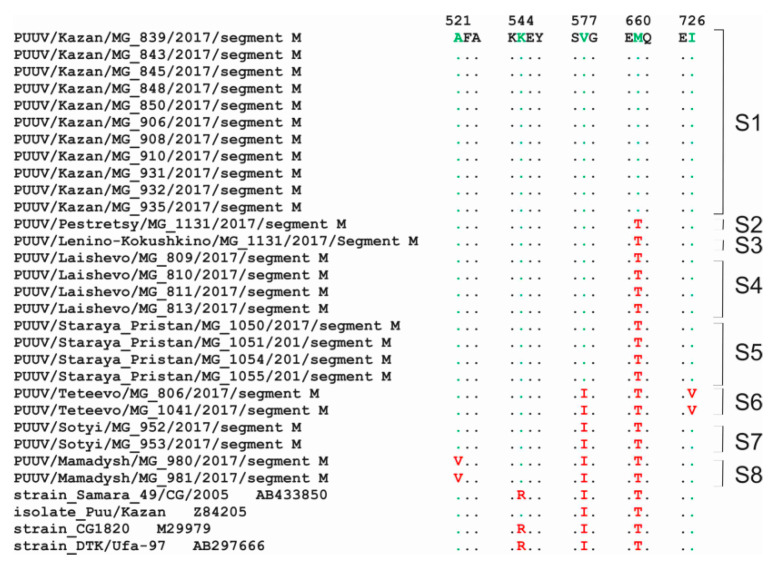
Mutations in the aa sequences of the Gn and Gc glycoprotein precursor.

**Figure 4 pathogens-09-00540-f004:**
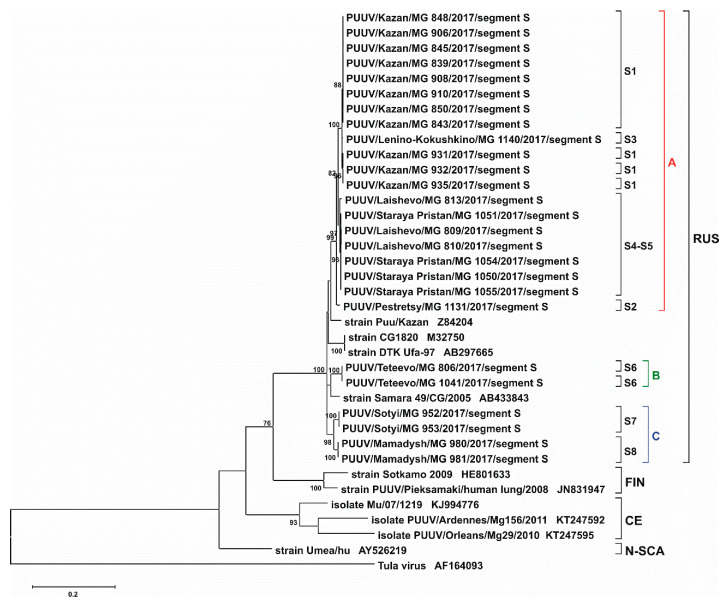
Phylogenetic tree for the partial S segment of “RT-2017” (nt 240-1296 based on GenBank sequence Z84204). Bootstrap values were calculated for 1000 replicates; only values greater than 70% are shown.

**Figure 5 pathogens-09-00540-f005:**
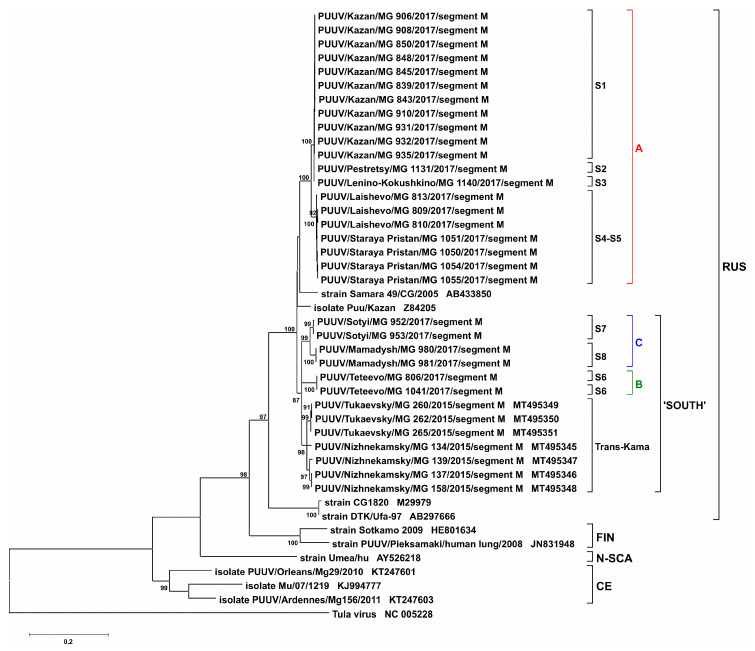
Phylogenetic tree for the partial M segment of “RT-2017” (nt 1499-2512 based on GenBank sequence Z84205). Bootstrap values were calculated for 1000 replicates; only values greater than 70% are shown.

**Figure 6 pathogens-09-00540-f006:**
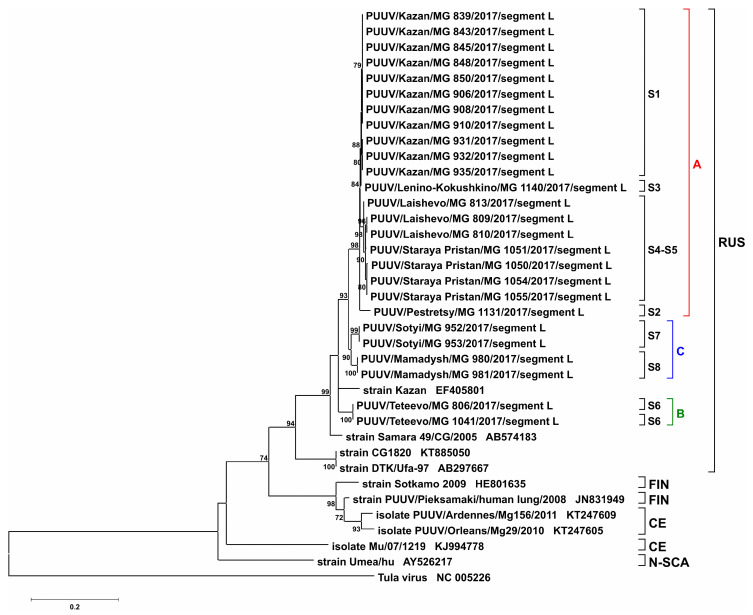
Phylogenetic tree for the partial L segment of “RT-2017” (nt 958-1622 based on GenBank sequence EF405801). Bootstrap values were calculated for 1000 replicates; only values greater than 70% are shown.

**Figure 7 pathogens-09-00540-f007:**
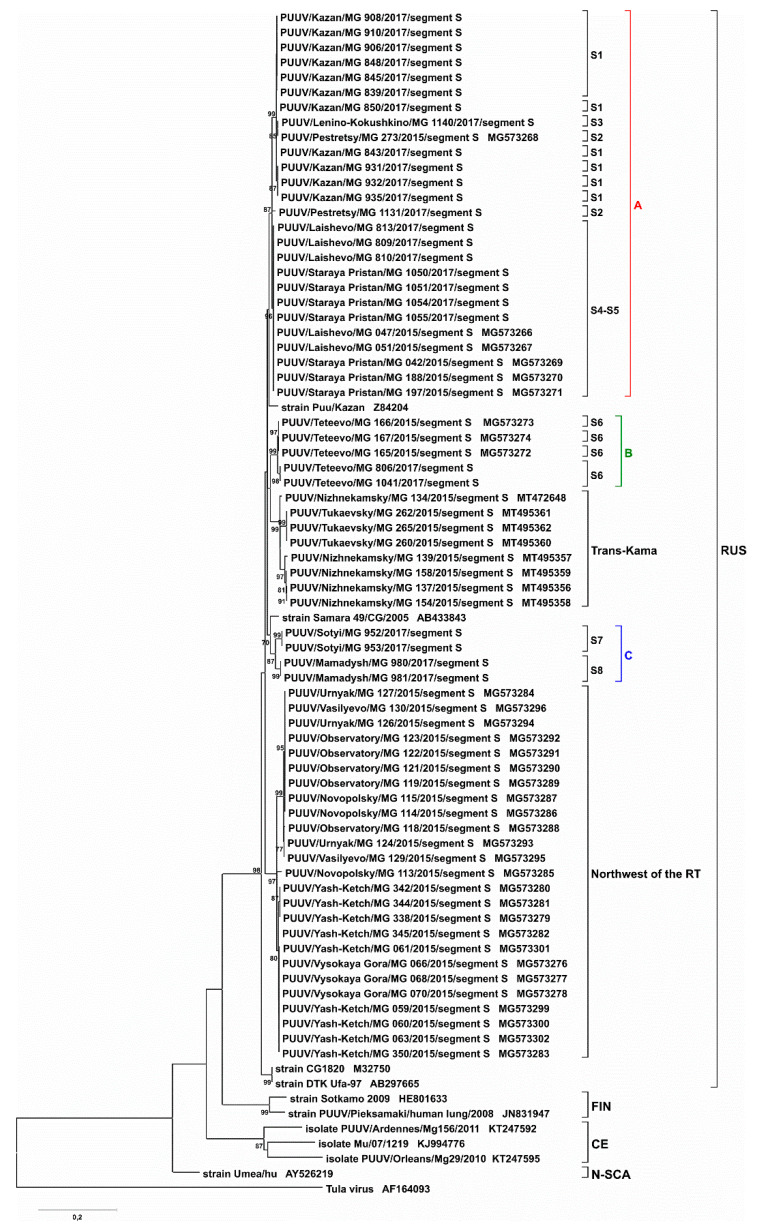
Phylogenetic tree for the partial S segment of PUUV strains from RT (nt 242–805 based on GenBank sequence Z84204). Bootstrap values were calculated for 1000 replicates; only values greater than 70% are shown.

**Table 1 pathogens-09-00540-t001:** List of locations, trapping sites, the number of trapped bank voles, reverse transcription-polymerase chain reaction (RT-PCR) screening results and the number of sequences used for analysis.

Location	Trapping Site	Number of Trapped Bank Voles	Number of RT-PCR Positive	No. of Sequences Used for Analysis ^a^		
				Segment S (nt 240–1296)	Segment M (nt 1499–2512)	Segment L (nt 958–1622)
Kazan	S1	46	12	11	11	11
Pestretsy	S2	4	1	1	1	1
Lenino-Kokushkino	S3	3	1	1	1	1
Laishevo	S4	6	4	3	3	3
Staraya Pristan’	S5	15	4	4	4	4
Teteevo	S6	29	2	2	2	2
Sotyi	S7	12	3	2	2	2
Mamadysh	S8	4	2	2	2	2
In total		119	29	26	26	26

^a^ Numbering corresponds to nucleotide sequences of PUUV strain Puu/Kazan (Genbank Z84204, Z84205 and, EF405801 for S, M and L segment, respectively).

**Table 2 pathogens-09-00540-t002:** Divergence (%) between “RT-2017” (Republic of Tatarstan) partial segment S nucleotide sequences and aa sequences (bold).

Cluster			A					B	C	
	Site	Direction	S1	S2	S3	S4	S5	S6	S7	S8
A	S1	nt →	0.0–0.6	2.3–2.4	0.6	1.9–2.1	1.8–2.0	6.0–6.1	6.0–6.2	5.5–5.8
		**aa ↓**	**0.0**							
	S2			0.0	2.4	1.7–2.0	1.8–1.9	5.5–5.6	5.9	5.1–5.2
			**0.6**	**0.0**						
	S3				0.0	2.0–2.1	1.9–2.0	6.1	6.2	5.7–5.8
			**0.0**	**0.6**	**0.0**					
	S4					0.0–0.8	0.1–0.8	5.4–5.8	5.6–5.9	5.1–5.5
			**0.6**	**0.0**	**0.6**	**0.0**				
	S5						0.0–0.2	5.2–5.3	5.4–5.5	4.9–5.1
			**0.6**	**0.0**	**0.6**	**0.0**	**0.0**			
B	S6							0.1	6.0–6.1	5.9–6.0
			**0.6**	**0.6**	**0.6**	**0.6**	**0.6**	**0.0**		
C	S7								0.0	2.2–2.3
			**0.6**	**0.0**	**0.6**	**0.0**	**0.0**	**0.6**	**0.0**	
	S8									0.1
			**0.6**	**0.0**	**0.6**	**0.0**	**0.0**	**0.6**	**0.0**	**0.0**

nt—nucleotide sequences; aa—amino acid sequences.

**Table 3 pathogens-09-00540-t003:** Divergence (%) between “RT-2017” partial segment S, M and L nucleotide sequences identified in different trapping sites and strains belonging to Russian (RUS), Finnish (FIN), Central European (CE) and north-Scandinavian (N-SCA) genetic lineages.

Trapping site	Genetic lineage									
	RUS				FIN		CE			N-SCA
	Name of Strain									
	Samara_49/CG/2005	Puu/Kazan	CG1820	DTK/Ufa-97	Sotkamo 2009	PUUV/Pieksamaki/human_lung/2008	Mu/07/1219	PUUV/Ardennes/Mg156/2011	PUUV/Orleans/Mg29/2010	Umea/hu
Segment S										
S1	6.4–6.5	5.1–5.3	6.3–6.6	6.2–6.5	17.6–17.7	16.4–16.5	19.7–19.9	19.2–19.6	21.2–21.3	18.4–18.7
S2	6.1	4.5	5,8	5.7	17.3	16.4	18.7	19.4	21.3	18.7
S3	6.5	5.3	6.6	6.5	18.0	16.7	19.8	19.7	21.3	18.7
S4	5.8–6.1	4.6–4.7	6.4–6.7	6.3–6.6	17.3–18.1	16.6–16.7	19.1–19.4	19.3–19.6	21.0–21.1	18.7–19.0
S5	5.9–6.1	4.4–4.5	6.2–6.4	6.1–6.3	17.8–18.1	16.5–16.7	19.3–19.4	19.2–19.3	20.9–21.0	18.7–19.0
S6	4.8–4.9	6.1–6.2	6.6–6.7	6.5–6.6	18.2	15.2–16.3	18.7	20.3–20.5	21.8–21.9	18.4
S7	4.8	5.4	6.6	6.5	17.1	15.4	19.5	19.2	21.3	19.5
S8	4.7–4.8	4.8–4.9	6.6–6.7	6.5–6.6	16.8	15.4	18.4	19.2	21.2	19.7–19.9
Segment M										
S1	6.6–6.7	6.3–6.6	16.0–16.1	15.3–15.4	18.0–18.4	18.9–19.4	23.1–23.4	21.6–22.0	25.1–25.2	22.0–22.2
S2	6.6	6.3	16.0	15.3	18.0	18.9	23.0	21.6	25.1	22.0
S3	6.6	6.3	15.7	15.0	18.0	18.9	23.1	21.6	24.8	22.0
S4	7.2–7.6	6.5–6.6	16.2–16.4	15.5–15.7	18.5–18.8	19.6–19.8	23.5–23.7	21.8–22.5	24.1–24.2	22.9–23.2
S5	7.6–7.7	6.7–6.8	16.2–16.6	15.5–15.9	18.4–18.8	19.6–20.0	23.8–24.1	22.2–22.3	24.4–24.8	23.0–23.2
S6	8.6–8.7	7.0–7.1	17.3	16.5	18.7	18.8–18.9	24.6–24.7	21.5–21.6	23.5–23.6	22.1–22.3
S7	7.3–7.4	6.5	14.6	13.9	17.7	18.1	24.3–24.4	21.2–21.3	23.6–23.8	21.0–21.3
S8	7.8	6.9	14.9	14.2	18.1	17.7	24.4	21.3	24.4	22.1
Segment L										
S1	7.4	7.2–7.5	14.5–14.9	14.5–14.9	17.6–18.0	18.4–18.8	20.7–21.1	21.0–21.5	20.2–20.6	22.7–23.2
S2	7.7	8.9	15.3	15.3	19.4	19.6	20.0	22.3	20.8	23.3
S3	7.4	7.5	14.9	14.9	18.0	18.4	21.1	21.0	20.2	23.2
S4	7.7–8.3	8.1–8.9	14.9–15.9	14.9–15.9	18.6–18.8	18.6–18.8	21.3	21.7–21.9	20.4–20.6	24.0–24.5
S5	8.1–8.6	8.4–8.9	15.3–15.9	15.3–15.9	18.4–18.64	18.4–18.6	21.3–21.9	21.5–21.7	20.2–20.6	24.3–24.7
S6	6.7	6.7	14.9	14.9	19.2	18.6	21.0	20.8	19.1	23.1
S7	7.6	7.4	14.5	14.5	19.4	18.8	22.1	21.0	20.2	22.9
S8	6.7	7.7	14.3	14.3	18.4	17.6	21.9	20.4	19.5	23.4

**Table 4 pathogens-09-00540-t004:** Divergence (%) between “RT-2017” partial segment M nucleotide sequences and aa sequences (bold).

Cluster			A					B	C	
	Site	Direction	S1	S2	S3	S4	S5	S6	S7	S8
A	S1	nt →	0.0–0.3	0.2–0.5	0.2–0.5	2.3–2.7	2.3–2.7	8.1–8.4	6.7–7.0	7.0–7.3
		**aa ↓**	**0.0**							
	S2			0.0	0.2	2.3–2.4	2.3–2.4	8.1–8.3	6.8–6.9	7.1
			**0.3**	**0.0**						
	S3				0.0	2.3–2.4	2.3–2.4	8.1–8.3	6.8–6.9	7.1
			**0.3**	**0.0**	**0.0**					
	S4					0.0–0.5	0.3–0.5	8.0–8.5	6.6–7.0	7.7–7.8
			**0.3**	**0.0**	**0.0**	**0.0**				
	S5						0.0–0.3	8.3–8.5	6.8–7.0	7.7–7.8
			**0.3**	**0.0**	**0.0**	**0.0**	**0.0**			
B	S6							0.1	5.8–6.1	6.2–6.4
			**0.9–1.2**	**0.6–0.9**	**0.6–0.9**	**0.6–0.9**	**0.6–0.9**	**0.3**		
C	S7								0.2	2.1–2.3
			**0.6**	**0.3**	**0.3**	**0.3**	**0.3**	**0.3–0.6**	**0.0**	
	S8									0.0
			**0.9**	**0.6**	**0.6**	**0.6**	**0.6**	**0.6–0.9**	**0.3**	**0.0**

**Table 5 pathogens-09-00540-t005:** Divergence (%) between “RT-2017” partial segment S, M and L aa sequences identified in different trapping sites and strains belonging to RUS, FIN, CE and N-SCA genetic lineages.

Trapping site	Genetic lineage									
	RUS				FIN		CE			N-SCA
	Name of Strain									
	Samara_49/CG/2005	Puu/Kazan	CG1820	DTK/Ufa-97	Sotkamo 2009	PUUV/Pieksamaki/human_lung/2008	Mu/07/ 1219	PUUV/Ardennes/Mg156/2011	PUUV/Orleans/Mg29/2010	Umea/hu
Segment S										
S1	0.6	1.1	1.4	1.1	3.5	3.2	2.6	3.5	4.4	3.2
S2	0.6	0.6	0.9	0.6	2.9	2.6	2.0	2.9	3.6	2.6
S3	0.6	1.1	1.4	1.1	3.5	3.2	2.6	3.5	4.4	3.2
S4	0.6	0.6	0.9	0.6	2.9	2.6	2.0	2.9	3.8	2.6
S5	0.6	0.6	0.9	0.6	2.9	2.6	2.0	2.9	3.8	2.6
S6	0.0	1.1	0.9	0.6	3.5	3.2	2.3	2.9	4.1	3.2
S7	0.6	0.6	0.9	0.6	2.9	2.6	2.0	2.9	3.8	2.6
S8	0.6	0.6	0.9	0.6	2.9	2.6	2.0	2.9	3.8	2.6
Segment M										
S1	1.2	0.9	3.6	2.1	3.6	3.6	9.1	7.5	8.1	9.1
S2	0.9	0.6	3.3	1.8	3.3	3.3	8.8	7.1	7.8	8.8
S3	0.9	0.6	3.3	1.8	3.3	3.3	8.8	7.1	7.8	8.8
S4	0.9	0.6	3.3	1.8	3.3	3.3	8.8	7.1	7.8	8.8
S5	0.9	0.6	3.3	1.8	3.3	3.3	8.8	7.1	7.8	8.8
S6	0.9–1.2	0.6–0.9	3.3–3.6	1.8–2.1	3.3–3.6	3.3–3.6	9.1–9.5	7.5–7.8	8.1–8.5	8.8–9.1
S7	0.6	0.3	3.0	1.5	3.0	3.0	8.8	7.1	7.8	8.5
S8	0.9	0.6	3.3	1.8	3.3	3.3	9.1	7.5	8.1	8.8
Segment L										
S1	0.9	0.5	1.4	1.4	6.1	6.6	9.2	8.1	7.1	10.7
S2	0.9	0.5	1.4	1.4	6.1	6.6	9.2	8.1	7.1	10.7
S3	0.9	0.5	1.4	1.4	6.1	6.6	9.2	8.1	7.1	10.7
S4	0.9	0.5	1.4	1.4	6.1	6.6	9.2	8.1	7.1	10.7
S5	0.9–1.4	0.5–0.9	1.4–1.8	1.4–1.8	5.6–6.1	6.1–6.6	9.2–9.7	7.6–8.1	6.6–7.1	10.7–11.2
S6	1.4	0.9	1.8	1.8	6.6	7.1	9.7	8.6	7.6	11.2
S7	0.9	0.5	1.4	1.4	6.1	6.6	9.2	8.1	7.1	10.7
S8	0.9	0.5	1.4	1.4	6.1	6.6	9.2	8.1	7.1	10.7

**Table 6 pathogens-09-00540-t006:** Divergence (%) between “RT-2017” partial segment L nucleotide sequences and aa sequences (bold).

Cluster			A					B	C	
	Site	Direction	S1	S2	S3	S4	S5	S6	S7	S8
A	S1	nt →	0.0–0.3	2.9	0.3	1.2–2.0	1.8–2.0	6.5	4.9–5.2	4.4
		**aa ↓**	**0.0**							
	S2			0.0	2.6	2.9–3.7	3.6–3.7	8.0	7.0	5.5
			**0.0**	**0.0**						
	S3				0.0	0.9–1.7	1.5–1.7	6.5	4.9	4.0
			**0.0**	**0.0**	**0.0**					
	S4					0.0–0.8	0.5–0.9	7.2–8.1	5.5–6.4	4.7–5.5
			**0.0**	**0.0**	**0.0**	**0.0**				
	S5						0.0–0.6	7.5–7.9	5.9–6.4	5.0–5.5
			**0.0–0.5**	**0.0–0.5**	**0.0–0.5**	**0.0–0.5**	**0.0–0.5**			
B	S6							0.0	6.7	6.7
			**0.5**	**0.5**	**0.5**	**0.5**	**0.5–0.9**	**0.0**		
C	S7								0.0	3.1
			**0.0**	**0.0**	**0.0**	**0.0–0.5**	**0.0–0.5**	**0.5**	**0.0**	
	S8									0.0
			**0.0**	**0.0**	**0.0**	**0.0–0.5**	**0.0–0.5**	**0.5**	**0.0**	**0.0**

**Table 7 pathogens-09-00540-t007:** Genotypes of “RT-2017” identified at various sites.

Cluster	Site	Strain	Segment ^a^		
			partial S (nt 240–1296)	partial M (nt 1499–2512)	partial L (nt 958–1622)
A	S1	839	A1	A1	A1
		843	A1	A1	A1
		845	A1	A1	A1
		848	A1	A1	A1
		850	A1	A1	A1
		906	A1	A1	A1
		908	A1	A1	A1
		910	A1	A1	A1
		931	A1	A1	A1
		932	A1	A1	A1
		935	A1	A1	A1
	S2	1131	A2	A1	A2
	S3	1140	A1	A1	A1
	S4	809	A3	A3	A3
		810	A3	A3	A3
		813	A3	A3	A3
	S5	1050	A3	A3	A3
		1051	A3	A3	A3
		1054	A3	A3	A3
		1055	A3	A3	A3
B	S6	806	B	B	B
		1041	B	B	B
C	S7	952	C1	C1	C1
		953	C1	C1	C1
	S8	980	C2	C2	C2
		981	C2	C2	C2

^a^ Numbering corresponds to nucleotide sequences of PUUV strain Puu/Kazan (Genbank Z84204, Z84205 and, EF405801 for S, M and L segment, respectively).

**Table 8 pathogens-09-00540-t008:** Primers used for RT-PCRs and sequencing analysis.

Genome Segment	Primer Name	Sequence (5′→3′)	Position ^a^ (nt)	Direction	Reference
S	PUUV-S-F1	tagtagtagactccttgaaaagc	1–23	Forward	This paper
	PUUV-S-F41	agctactacgagaacaactgg	21–41	Forward	This paper
	PuuV-for	ctgcaagccaggcaacaaacagtgtcagca	172–201	Forward	[9]
	4S-F3	gcactggaggataaactcgc	199–218	Forward	This paper
	PUUV-S-F704	aacatcatgagtccagtaatggg	682–704	Forward	This paper
	69S-F3	ttatggcatctaaaactgtgg	1079–1099	Forward	This paper
	PUUV-S-R1496	gtataattccagttaacccctg	1496–1517	Reverse	This paper
	PUUV-S-R1183	gtacagtaggattatcctctgatc	1183–1206	Reverse	This paper
	PuuV-Rev	gtctgccacatgatttttgtcaagcacatc	865–894	Reverse	[9]
	5S-B3	ggccagtctttaagcaagaaag	719–740	Reverse	This paper
	R358 PUUVS	catttacatcaaggacatttcc	337–358	Reverse	This paper
M	F1452 PUUVM	tctttaatcccaggagttgc	1451–1470	Reverse	[16]
	R2582 PUUVM	aaattgtccctattaaacacac	2561–2582	Reverse	[16]
L	F925-PUUL	agcttccaagcaccatatttaccatc	925–950	Forward	This paper
	R1663-PUUL	gattatctgcatcaataagacctagt	1638–1663	Reverse	This paper

^a^ Numbering corresponds to nucleotide sequences of PUUV strain Puu/Kazan (Genbank Z84204, Z84205 and, EF405801 for S, M and L segment, respectively).

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
