# Peer review of "Prevalence of the Puumala orthohantavirus Strains in the Pre-Kama Area of the Republic of Tatarstan, Russia"

_pathogens, 2020, doi:10.3390/pathogens9070540_

Round 1

Reviewer 1 Report

In their article „Prevalence of the Puumala orthohantavirus strains in the Pre-Kama area of the Republic of Tatarstan, Russia” Davidyuk et al. aim at analysing the genetic variability of Puumala virus (PUUV) in the Pre-Kama region in Russia. PUUV sequences were obtained from bank voles trapped in 2017. According to their phylogenetic analysis of all three genomic segments, genetic distance does not directly correlate with geographical distance, but is rather due to several re-colonisation events of the bank voles into the Pre-Kama region after the last ice-age. Differing placement of one strain in the M-segment based phylogenetic tree compared to the S and M phylogenies leads them to conclude a reassortment event.

General remarks

The authors provide a comprehensive description of their dataset and compare it with other PUUV strains belonging to the Russian lineage. Their conclusions are based on phylogenetic trees which were produced using the Maxium Parsimony (MP) method. For several reasons, MP is not considered state of the art for producing phylogenetic trees that describe the “true” evolutionary history of sequences, for example, it does not take into account any model of nucleotide evolution. As the authors use the resulting trees for extensive interpretations, they should use Maximum Likelihood- or Bayesian-based methods to infer the phylogenetic history of the presented sequences. Short branch lengths provide only limited information for the interpretation of phylogenetic trees. Using multiple inferring algorithms, different models of nucleotide evolution, and various rooting methods would show the robustness of the results and increase the confidence in the drawn conclusions.

The data itself is worth being published, however, the methods need to be improved. Since interpretation of the data might change with new analyses, I will not comment on the interpretation in this review. The vulnerability of the analysis can be seen when comparing Figures 4 and 7. Both analyses are based on the S-segment, however, the placement of strain MG1131/2017 differs remarkably.

Specific remarks

Regardless of the method used, the authors should provide information on the rooting method used or whether the phylogenetic trees are unrooted as well as the basis of the given branch support values (see Szabo et al, 2017).

The M-segment based tree contains sequences from the Trans-Kama area which are missing in the L- and S-based tree. For direct comparison of phylogenies of the different segments, it would be preferable to show the same strains in each tree (fig 4,5,6).

Table 1: It would be helpful to have a distance measure on the map

Line 35: typo (analysed)

Line 35-37: typo (correlated)

Line 92: give % of positive tested bank voles

Line 102-103: logic? Probably a “no” missing

Line 172: It is not clear what the numbers refer to. Probably clades A, B, C?

Lines 175-176: it might be easier for the reader if the authors gave the sample site description together with the strain name (S2 and S3).

Line 204: “a reassortant” instead of “the”

Author Response

General remarks

The authors provide a comprehensive description of their dataset and compare it with other PUUV strains belonging to the Russian lineage. Their conclusions are based on phylogenetic trees which were produced using the Maxium Parsimony (MP) method. For several reasons, MP is not considered state of the art for producing phylogenetic trees that describe the “true” evolutionary history of sequences, for example, it does not take into account any model of nucleotide evolution. As the authors use the resulting trees for extensive interpretations, they should use Maximum Likelihood- or Bayesian-based methods to infer the phylogenetic history of the presented sequences. Short branch lengths provide only limited information for the interpretation of phylogenetic trees. Using multiple inferring algorithms, different models of nucleotide evolution, and various rooting methods would show the robustness of the results and increase the confidence in the drawn conclusions.

The data itself is worth being published, however, the methods need to be improved. Since interpretation of the data might change with new analyses, I will not comment on the interpretation in this review. The vulnerability of the analysis can be seen when comparing Figures 4 and 7. Both analyses are based on the S-segment, however, the placement of strain MG1131/2017 differs remarkably.

Answer.

Agree: To construct phylogenetic trees, we used both Maximum Parsimony and Maximum Likelihood methods. The topology of the trees constructed using these methods was almost identical. Thus, in this case, both methods were equivalent and did not influence the interpretation of the data. The trees constructed using the MP method seemed to be more convenient for visual perception and that’s why we used them in the manuscript. However, according to your recommendations, we made changes by replacing the MP trees with ML trees in the manuscript  and moved the MP-trees to the Supplementary.

Specific remarks

Regardless of the method used, the authors should provide information on the rooting method used or whether the phylogenetic trees are unrooted as well as the basis of the given branch support values (see Szabo et al, 2017).

The M-segment based tree contains sequences from the Trans-Kama area which are missing in the L- and S-based tree. For direct comparison of phylogenies of the different segments, it would be preferable to show the same strains in each tree (fig 4,5,6).

Answer.

Agree: the information about methods was added. The S segment sequences of PUUV strains from Trans-Kama were included to the tree (Fig. 7). The phylogenetic tree for L segment was not modified because other sequences for L segment of PUUV strains in the RT were not obtained. All phylogenetic trees were checked and corrected.

Table 1: It would be helpful to have a distance measure on the map

Answer.

Agree: the distance measure was added on the map.

Line 35: typo (analysed)

Answer.

Agree: the typo was corrected.

Line 35-37: typo (correlated)

Answer.

Agree: the typo was corrected.

Line 92: give % of positive tested bank voles

Answer.

Agree: the % of positive tested bank voles was added to the text.

Line 102-103: logic? Probably a “no” missing

Answer.

Agree: the word ‘no’ was added to the text.

Line 172: It is not clear what the numbers refer to. Probably clades A, B, C?

Answer.

Agree: numbers were changed to the clade’s designation (A, B, C).

Lines 175-176: it might be easier for the reader if the authors gave the sample site description together with the strain name (S2 and S3).

Answer.

Agree: the trapping site numbers were added.

Line 204: “a reassortant” instead of “the”

Answer.

Agree: the article was corrected.

Reviewer 2 Report

General comments:

In “Prevalence of the Puumala orthohantavirus Strains in the Pre-Kama Area of the Republic of Tatarstan, Russia” Davidjuk et al. perform phylogenetic analysis of 26-PUUV sequences obtained from bank voles captured at seven sites in the Pre-Kama region.

Main comments:

This manuscript describes a detailed phylogenetic analysis of the PUUV sequences obtained. The conclusions and discussion of the results are interesting, thorough, and well thought out. My comments are mainly minor and listed below.

Also, I think there may be an issue with the formatting of Tables 2, 4, and 6. Please check carefully these are correct.

Minor comments:

Line 92 – I know this information is in the M&M, but I think it should also be stated here that the PUUV RNA was detected in the lung tissue of these animals.

Line 96 – I think this figure would benefit from having a globe or something similar next to the current image showing where this area is in Russia/world. Also, please expand on the figure legend to fully describe the map, what are the black regions? What is the black line on the R. Volga?

Line 103 – I think this should read “no significant differences”?

Line 114 – The first two rows of this table are very difficult to interpret, trying to work out what strains (row 2) are linked to the location (row 1). Consider either breaking up the line between row 1 and 2, or adding vertical columns to separate out the location.

Line 136 – same comment as Line 144 for table clarity

Line 150 – same comment as Line 144 for table clarity

Line 174 – I don’t know what is meant by department?

Line 195 – I think this should read reassortant, or something similar?

Author Response

Main comments:

This manuscript describes a detailed phylogenetic analysis of the PUUV sequences obtained. The conclusions and discussion of the results are interesting, thorough, and well thought out. My comments are mainly minor and listed below.

Also, I think there may be an issue with the formatting of Tables 2, 4, and 6. Please check carefully these are correct.

Answer.

Agree: the tables were corrected.

Minor comments:

Line 92 – I know this information is in the M&M, but I think it should also be stated here that the PUUV RNA was detected in the lung tissue of these animals.

Answer.

Agree: the information was added.

Line 96 – I think this figure would benefit from having a globe or something similar next to the current image showing where this area is in Russia/world. Also, please expand on the figure legend to fully describe the map, what are the black regions? What is the black line on the R. Volga?

Answer.

Agree: Fig. 1 was corrected according to the reviewer's comment.

Line 103 – I think this should read “no significant differences”?

Answer.

Agree: the word ‘no’ was added to the text.

Line 114 – The first two rows of this table are very difficult to interpret, trying to work out what strains (row 2) are linked to the location (row 1). Consider either breaking up the line between row 1 and 2, or adding vertical columns to separate out the location.

Answer.

Agree: the table was corrected.

Line 136 – same comment as Line 144 for table clarity

Answer.

Agree: the table was corrected.

Line 150 – same comment as Line 144 for table clarity

Answer.

Agree: the table was corrected.

Line 174 – I don’t know what is meant by department?

Answer.

Agree: it was corrected and replaced by more suitable word “branch”.

Line 195 – I think this should read reassortant, or something similar?

Answer.

We used the expression ‘genome variant’ because more analysis, presented in Table 7, was done  to confirm that one of the genome variants (SA2-MA1-LA2) was reassortant.

Round 2

Reviewer 1 Report

The authors well addressed the concerns regarding the methodology and described it sufficiently in the material and methods part. Please also to add the relevant information into the caption of the figures in the main text, according to how it was done in the supplementary information.

There are two minor additions that could be added to improve the manuscript:
For the genotypes stated in table 7, did the authors define a general “cut-off” for the different genotypes? It would be nice to have that information.

Line 221: Please add a reference on PUUV reassortment (e.g. Szabo et al 2017)

Author Response

The authors well addressed the concerns regarding the methodology and described it sufficiently in the material and methods part. Please also to add the relevant information into the caption of the figures in the main text, according to how it was done in the supplementary information.

Agree: the information was added to captions.

There are two minor additions that could be added to improve the manuscript:
For the genotypes stated in table 7, did the authors define a general “cut-off” for the different genotypes? It would be nice to have that information.

Agree: the required information was added to table 7.

Line 221: Please add a reference on PUUV reassortment (e.g. Szabo et al 2017)

Agree: the reference was added to the text (lines 233-235).